# Rise and Recharge: Exploring Employee Perceptions of and Contextual Factors Influencing an Individual-Level E-Health Smartphone Intervention to Reduce Office Workers’ Sedentary Time at Work

**DOI:** 10.3390/ijerph18189627

**Published:** 2021-09-13

**Authors:** Abigail S. Morris, Kelly A. Mackintosh, Neville Owen, Paddy C. Dempsey, David W. Dunstan, Melitta A. McNarry

**Affiliations:** 1School of Sport and Exercise Sciences, Swansea University, Swansea SA1 8EN, UK; a.morris7@lancaster.ac.uk (A.S.M.); k.mackintosh@swansea.ac.uk (K.A.M.); 2Department of Health Research, Faculty of Health and Medicine, Lancaster University, Lancaster LA1 4YW, UK; 3Baker Heart & Diabetes Institute, Melbourne, VIC 3004, Australia; neville.owen@baker.edu.au (N.O.); Paddy.Dempsey@mrc-epid.cam.ac.uk (P.C.D.); david.dunstan@baker.edu.au (D.W.D.); 4MRC Epidemiology Unit, Cambridge Biomedical Campus, Institute of Metabolic Science, University of Cambridge, Cambridge CB2 0SL, UK; 5Diabetes Research Centre, University of Leicester, Leicester General Hospital, Leicester LE5 4PW, UK

**Keywords:** feasibility, workplace, intervention, sedentary behavior, physical activity, standing, mobile application, EMA

## Abstract

This feasibility study explored the contextual factors influencing office workers’ adherence to an e-health intervention targeting total and prolonged sedentary time over 12 weeks. A three-arm quasi-randomized intervention included prompts at 30 or 60 min intervals delivered via a smartphone application, and a no-prompt comparison arm. Fifty-six office workers completed baseline (64% female) and 44 completed the 12 week follow-up (80% retention). Ecological momentary assessments (EMA) captured contextual data, with 82.8 ± 24.9 EMA prompt questionnaires completed weekly. Two focus groups with *n* = 8 Prompt 30 and 60 participants were conducted one-month post-intervention to address intervention acceptability and feasibility. Contextual findings indicate that when working on a sedentary task (i.e., reading or screen-based work) and located at an individual workstation, hourly prompts may be more acceptable and feasible for promoting a reduction in total and prolonged sedentary time compared to 30 min prompts. Interpersonal support also appears important for promoting subtle shifts in sedentary working practices. This novel study gives a real-time insight into the factors influencing adherence to e-health prompts. Findings identified unique, pragmatic considerations for delivering a workplace e-health intervention, indicating that further research is warranted to optimize the method of intervention delivery prior to evaluation of a large-scale intervention.

## 1. Introduction

Occupational sedentary time is the largest contributor to total weekday sedentary time among office workers [1], with employed adults in the UK typically spending between 60–80% of their working hours sedentary [2]. Of particular concern is that desk-based workers in the UK spend over half of their sedentary time in prolonged bouts of either ≥30 or ≥55 min [3]. This protracted volume of total and prolonged sedentary behavior (SB) is associated with adverse health, well-being and work outcomes, including increased cardiometabolic risk factors [4,5], musculoskeletal discomfort [6,7], impaired cerebral blood flow [8] and lower productivity [9]. Given the high exposure of office and desk-based workers to these potential health risks, there is an urgent need to target sedentary working patterns by implementing frequent breaks from prolonged SB and increased opportunities for physical activity (PA) [10,11].

Workplace interventions involving height-adjustable workstations, with or without the addition of information and counselling, have been shown to be effective for reducing total and prolonged occupational sedentary time among desk-based workers over three months (−84 to −116 min per workday [12]), with observed reductions maintained over 12-month follow-ups [13,14]. Implementing such intervention strategies can, however, be costly. It is therefore unsurprising that a return on investment is commonly perceived as a barrier to wide-scale implementation of standing desk-based approaches to reducing prolonged SB [15,16,17,18]. Electronic-health (e-health) interventions, which encompass “the development and use of digital technologies to improve health” (p. 5, [19]), may offer an alternative approach. E-health strategies, particularly those involving wearable activity trackers and smartphone applications, are one of the most rapidly growing market sectors and allow behavioral interventions to be delivered in real time, across a variety of settings [18,20,21]. To date, workplace e-health interventions primarily utilize prompts, cues and self-monitoring as behavior change techniques [22]. Preliminary evidence from a recent systematic review suggests that workplace interventions consisting of a smartphone application and wearable technology can elicit short-term (3 month) reductions in total daily SB of an average of 42.4 min/workday [23]. Given that a high proportion (80%) of the working-age population in the UK utilize a smartphone device daily, this highlights the potential for e-health technologies to provide a low-cost and widely accessible alternative to address what has been described as the pandemic of sedentary behavior [21,22,24].

Compared to multi-component workplace interventions involving height-adjustable workstations, the effectiveness of workplace computer, mobile and wearable technology interventions appears to decrease over time [22]. Indeed, user engagement in a variety of e-health interventions typically declines during the first few weeks of delivery [20]. Understanding the factors influencing participant engagement is therefore crucial since intervention dose is associated with intervention efficacy [20]. Despite the importance of understanding the factors underpinning this relatively rapid decline in e-health intervention effectiveness, no studies have explored the contextual factors influencing adherence to smartphone prompts targeting total and prolonged sedentary bouts, and importantly, how they may change over time [25]. Such findings would contribute significantly to the evidence base by providing a comprehensive, contextual understanding of the potential for e-health smartphone applications to be implemented as either a stand-alone or as an addition to multi-component workplace SB interventions [22].

The aim of this feasibility study therefore was to ascertain the contextual factors that influence the adherence to an e-health smartphone-based intervention targeting total and prolonged sedentary time during waking hours in office workers. Understanding the feasibility of a technology-based SB workplace intervention and the contextual factors influencing adherence to e-health prompts aims to support the development of future effective and sustainable workplace behavior change interventions [17,26].

## 2. Materials and Methods

### 2.1. Study Design

This study and the intervention have been described in full elsewhere [27]. Briefly, 56 office workers from three companies were quasi-randomized to one of two intervention arms or a no-prompt comparison arm. Those in the intervention arms received prompts via their smartphone to break up their sedentary time every 30 or 60 min (Prompt 30 or Prompt 60, respectively) during working hours for 12 weeks. This study was approved by Swansea University (2018-147).

### 2.2. Data Collection

In line with the primary aim to ascertain the contextual factors that moderate the adherence to an e-health smartphone application targeting office workers’ total and prolonged SB, Prompt 30 and Prompt 60 participants completed a series of ecological momentary assessment questions throughout the 12 week trial to capture real-time insights. Furthermore, at baseline, participants were asked to provide a description of their workstation set up and all participants were invited to a 1 h focus group a minimum of one month following completion of the 12 week trial.

### 2.3. Outcomes

#### 2.3.1. Ecological Momentary Assessment

To explore whether psychosocial or contextual (i.e., individual, interpersonal, environmental and organizational) factors influenced adherence to the 30 or 60 min prompts during work time, intervention participants received six ecological momentary assessment (EMA) prompt questionnaires per week via email for the duration of the 12 week intervention (Symlconnect Ltd., Swansea, UK) (Appendix A: Ecological Momentary Assessment questionnaires). These consisted of five items which were sent at random times during work hours only, to capture ‘real-time’ potential contextual influences on occupational SB [28,29]. Prompt questionnaires asked participants to recall what they were doing before the last prompt they received (job task), who they were with (interaction), where they were (location) and how they felt (affect). In addition, intervention participants received one weekly review form via email consisting of four, ten-point Likert scales to assess: how frequently they broke up their sedentary time; how often they broke their sedentary time in relation to the previous week; how likely they were to break up their sedentary time the following week; and how helpful they found the application for encouraging them to break up their occupational sedentary time. Response scales ranged from 0 (not at all) to 10 (all of the time), with five free-text questions to allow participants to provide further context and insight into their answers.

#### 2.3.2. Focus Groups

Prompt 30 and Prompt 60 participants were invited to attend a focus group on following completion of the 12 week intervention. Focus groups were conducted in December 2019. To elicit in-depth insights into the acceptability and feasibility of the intervention and the contextual factors influencing their adherence to the smartphone prompts during working hours, discussion areas included participant experiences and perspectives of using the smartphone application, motivations for participation, frequency of breaks, reasons for using the delay function, perceived impact (if any) on health, well-being and productivity outcomes and factors influencing maintenance of the smartphone application beyond completion of the trial. The first author (ASM, who is experienced in qualitative research) developed the focus group schedule, which was reviewed by members of the research team (MAM, KAM; Appendix A: Rise and Recharge focus group schedule). The protocol for focus group delivery was standardized by using the semi-structured schedule to promote a level of commonality across the focus groups [30], while allowing flexibility in the order and sequence of questions to encourage participants to respond openly and freely. Probes were used to elicit depth or clarity from responses [31]. On-the-spot member checking was used to establish interpretation and meaning during focus groups [32].

### 2.4. Analysis

#### 2.4.1. Ecological Momentary Assessment

Prompt forms and weekly review forms were collated and downloaded from Syml.connect. Fidelity to the EMA prompts and weekly reminders was calculated according to the number of participants who successfully completed a prompt (6 prompt forms per week) or review form (1 review form per week). Participant responses to the 5-item prompt form were collated and grouped by task, interaction, location and affect when the prompt went off. Affect was coded according to the eight affect categories identified in Watson, Wiese [33]. Due to low compliance to the EMA prompt and review questionnaires per week, data were analyzed using complete case analysis to determine participant’s mean response over 12 weeks [34].

#### 2.4.2. Focus Groups

Focus groups were audio recorded, transcribed verbatim and participants were assigned their unique participant ID to ensure responses were anonymized throughout. A descriptive thematic analysis was conducted to explore patterns and themes across the complete dataset in relation to the contextual factors influencing adherence to breaks [35,36]. Following each focus group, the first author reflected on participant responses and initial thoughts and patterns in a reflective commentary, which was then expanded upon during familiarization and the initial coding stages. The socio-ecological model provided the point of departure for the deductive analysis [37]. Sub-themes were generated during an inductive process and provided a structure and rich context to the pre-defined contextual categories [37]. The initial coding framework was presented by the first author (ASM) and reviewed by other authors (KAM and MAM; Appendix A: Thematic analysis framework). This triangulation process helped refine the themes and is considered an important step for ensuring trustworthiness in qualitative analysis [32]. Findings are reported in line with the Consolidated Criteria for Reporting Qualitative Research (COREQ) checklist [38].

#### 2.4.3. Statistical Analysis

Descriptive statistics were conducted on the complete EMA dataset. Due to the categorical nature of the Prompt and Review data, Chi-square tests were conducted using SPSS (IBM SPSS Statistics 26, Armonk, NY, USA) to explore associations between participant adherence and influencing contextual factors, with alpha set to *p* ≤ 0.05 [39]. Furthermore, based on a complete case analysis including all data collected across the 12 weeks, Likert scales from the weekly review forms were analyzed using the Mann–Whitney U test to determine differences between Prompt 30 and Prompt 60 arms [39].

## 3. Results

Overall, three organizations were recruited and 56 participants consented, completed baseline assessment and were assigned to either the no-prompt comparison (*n* = 21), Prompt 30 (*n* = 20) or Prompt 60 arms (*n* = 15). Attrition was 7% (no-prompt comparison *n* = 1, 5%; Prompt 30 *n* = 3, 15%; Prompt 60 *n* = 0, 0%) and 14% (no-prompt comparison *n* = 4, 27%; Prompt 30 *n* = 2, 13%; Prompt 60 *n* = 0, 0%) at 6 and 12 weeks, respectively, with 80% (no-prompt comparison *n* = 15, 71%; Prompt 30 *n* = 15, 75%; Prompt 60 *n* = 15, 100%) of total participants retained throughout the trial. All Prompt 30 and 60 participants were invited to take part in a one-off focus group, with *n* = 8 expressing an interest and able to attend. There were no withdrawals due to adverse events (Appendix A: Consort flow diagram of enrolment, allocation, follow-up and analyses).

### 3.1. Baseline Characteristics

Participants were predominantly female (64%), White British, full-time employees, educated to tertiary level with ≥3 years tenure. At baseline, 29 participants provided a description of their workstation set up. Of these, 93% reported working at a standard seated desk and 7% had access to a height-adjustable workstation. The majority (90%) had an individual workstation and access to an adjustable office chair, 72% worked in a shared office at a bank of desks, and 14% had access to additional ergonomic aids such as a foot stool and a back support. At baseline, participants were typically pre-hypertensive, overweight, had an elevated waist circumference, were sedentary for >10 h per day and spent 69% of work hours sedentary, 24% standing and 7% stepping.

#### 3.1.1. Ecological Momentary Assessment

Descriptive statistics to assess fidelity to the prompting forms identified that 994 of 2520 forms were completed (39%) across the trial. Between groups, 75% of Prompt 30 participants completed an average of 46.1 ± 29.1% and 93% of Prompt 60 participants completed an average of 48.7 ± 25.7% of the prompting forms over the 12 week intervention.

Of the weekly review forms, 143 of 420 forms were completed (34%). Between groups, 95% of Prompt 30 participants completed an average of 42.9 ± 26.3% and 80% Prompt 60 participants completed an average of 52.8 ± 28.5% of the weekly review forms. Participant compliance (calculated as the total number of participant responses) to both the prompt and review questionnaires declined by 55.5 and 66.6%, respectively over the 12 week intervention (Appendix A: Completed Prompt and Review forms over the 12 week intervention).

Overall, participants reported that they were mostly engaged in reading/desk-based work when prompted to break up their sedentary time by the application (80.3%), with participants typically located at their desk (63.2%) or in their office but not at their desk (16.5%). There were no differences between groups for main job task and location. A significantly higher proportion of Prompt 60 participants across the 12 week trial (weeks 1,2,4,5 and 7) were not alone when they received a prompt [X^2^ range *p* = 0.000–0.042], and were usually with a work colleague [X^2^ range *p* = 0.000–0.015], compared to Prompt 30 participants who were more commonly alone when they received a prompt. 

On average, the majority of participants reported feelings of pleasantness (42.3%) during the trial when the prompt went off, including feelings of contentment, happiness and satisfaction, with 10.6% of participants indicating strong engagement with their work. A high proportion of respondents, however, reported feelings of a low positive affect (16.4%), such as feeling sleepy, sluggish or drowsy, and high negative affect (13.0%), such as feeling anxious, stressed or hostile. Chi-square analysis of the categorical data from the prompt and review questionnaires shows a significant difference in affect for participants who were with a colleague when prompted [Prompt 30; *p* = 0.008; Prompt 60; *p* = 0.003], with a higher proportion of Prompt 30 participants reporting low positive and high negative affect and a higher proportion of Prompt 60 participants reporting pleasantness, strong engagement, high negative and a low positive affect compared to those who were alone (Table 1).

Results from the weekly review questionnaire Likert data are presented in Table 2. Likert data were not normally distributed, and therefore non-parametric analyses were conducted. Overall, distributions of Likert scores for Prompt 30 and Prompt 60 arms were similar, and medians scores across groups were not statistically significant [40].

Descriptive statistics indicated that the main factors negatively influencing participant engagement in the prompts included high workload (27.3%), meetings (18.9%) and being away from their desk or mobile (10.5%). Conversely, the e-health smartphone prompts (11.3%) and an intrinsic health-driven motivation (11.9%) were reported as positive factors influencing adherence. Chi square analysis of the categorical data indicated that there were no statistically significant associations between interventions arms and review questionnaire responses across the 12 week trial [X^2^ range *p* = 0.82–0.858].

#### 3.1.2. Focus Groups

Verbatim quotations are presented as participant number (P1-56), Company (Company 1 [C1] or Company 2 [C2]) and intervention arm (Prompt 30 or Prompt 60, e.g., P06 C1 Prompt 30). Ten participants, five (*n* = 4 male) from C1 and five (*n* = 1 male) from C2, took part. Overall, there were eight Prompt 30 participants and two Prompt 60 participants (Appendix A: Consort flow diagram of enrolment, allocation, follow up and analysis), with the mean focus group duration being 40.6 ± 2.0 min. Company 3 did not have any Prompt 30 or Prompt 60 participants and therefore did not contribute to the focus groups.

Focus group participants commonly appeared motivated to take part in the intervention by health-related factors, such as increased health awareness, health check feedback and the potential beneficial impact on health and well-being.


*“[…] So I was just curious as to how the intervention may help me actually sort of move around a bit more and what impact that may or may not have on my day to day work and my health as well.”*
P04 C1 Prompt 30


*“I wanted to get involved for selfish reasons really to find out my sort of cholesterol levels and blood sugar, blood glucose, because there is a history of heart disease and diabetes in my family so I thought it’ll be a good opportunity to find out where I stand with that and whether breaking up my sitting patterns in work would, you know, indeed help with that as well.”*
P51 C1 Prompt 60

The key contextual factors supporting fidelity to the 30 and 60 min prompts appeared consistent between arms and across C1 and C2 and predominantly included interpersonal support and cultural factors. Participants in C1 and C2 who worked in an office with multiple staff members reported that peer support encouraged both initial participation and ongoing engagement throughout the trial.


*“At the beginning you know I quite often when I saw other people on the team standing up, move around and do this that and the other, I thought oh this could be quite a good idea, because sometimes you can sit there for three hours.”*
P55 C1 Prompt 30

Participants in both companies reported that synchronized prompts among members within their teams enhanced peer support throughout the trial and was an effective strategy for prompting breaks to their sedentary time.


*“Me and C definitely did like I said, we work really close together so we could visually see each other. We were on the same 30 min [prompt] so it did feel like we were prompting each other.”*
P44 C2 Prompt 30


*“I think the other interesting thing about it, is there was quite a few of us in the office doing it [the prompts] often aligned.”*
P55 C1 Prompt 30

Enhanced interpersonal support throughout the trial was ascribed to an increased health awareness among colleagues in C1 and C2, resulting in a subtle, but noticeable, shift in working cultures over the 12 weeks. This shift facilitated actions such as breaking up sedentary time during informal meetings and standing while on video calls to be more commonly perceived as acceptable working behaviors.


*“I think [the intervention has] made [breaking up sedentary time] less of an odd behavior. Before, if someone stood up you’d think ‘oh why is he stood up to do his skype call’ whereas now maybe the whole idea of standing up at your desk is not that weird. So it’s acceptable.”*
P55 C1 Prompt 30


*“I’ve noticed people do stand-up more in the office when doing skype calls and things.”*
P51 C1 Prompt 60

In contrast to the individual and interpersonal factors facilitating breaks to sedentary time, the main factors influencing adherence to the prompts appeared to be organizational and environmental, including workload, job tasks, location and perceived organizational expectations. Commonly, factors negating participant engagement with the prompts were identified as high workload, particularly relating to tasks requiring high levels of concentration and focus. 


*“I did have to turn [my prompts] off for three weeks because my work just couldn’t let me take time to stand up.”*
P42 C2 Prompt 30


*“Quite a lot of the time I’d just press no because I’m busy”*
P55 C1 Prompt 30

Participants in C2 identified fluctuations between predominately seated, desk-based activities and job tasks involving regular breaks to sedentary time, or prolonged periods away from their workstation within their role, i.e., for meetings or site visits. This meant the prompts were more appropriate when working on predominately desk-based activities.


*“[During the intervention period] I was up and down up and down all the time. But now, at this particular point in time I’ve got a lot more desk work to do. And so now, I would be aware of having to get up because you know I know I’ve been sitting”*
P48 C2 Prompt 30

Similarly, a common reason cited for missing prompts was attributed to task variation. Participants reported that tasks involving being away from their desk and/or not being in close proximity to their mobile phone would result in missed prompts. Missing prompts therefore did not necessarily mean participants were engaged in sedentary working tasks but rather that it was not always practical, feasible or even appropriate to carry a mobile phone while at work.


*“[…] ironically the days I probably had my worst percentage score in terms of hitting ‘Yes, I’m up’ were my most active days. Because if I was out on site for two or three hours I wouldn’t pay attention to the phone. So I’d miss maybe six or seven consecutive prompts in which time I could’ve racked up maybe 6,7,8 thousand steps ”*
P05 C1 Prompt 30


*“I kept missing [the prompt] because I wasn’t my desk, didn’t have my phone on me and I’d get back and look at it and think, Oh, I missed two this time […] But it wasn’t that I was sitting anyway, I was up and down up and down all the time”*
P48 C2 Prompt 30

Indeed, when seated at their desk, participants commonly perceived the 30 and 60 min prompts to break up sedentary time as an acceptable reminder.


*“When I was [at my desk], you know, sit on my desk was fine, this wasn’t a problem that didn’t have an issue with it at all, was simple enough to use Yeah, it’s just quick press a button stand up”*
P39 C2 Prompt 30


*“[…] if I had a deadline, I was concentrating, if I had to something done then 60 min could just got like that and so I could be sitting close to two hours so the 60 min actually prompted me to move so I thought that was beneficial.”*
P05 C1 Prompt 60

Participants in Prompt 30 arm reported, however, that the frequency of the prompts was less feasible and that the prompts caused disruption to concentration, workflow and, in some instances, enhanced feelings of stress.


*“Those 30 min go really quickly. Yeah, you know, and although [breaking up your sedentary time is] very simple to do, when you know it is just like that but when you are kind of working and focusing, and times going quick you just think. Yeah, you kind of have a little bit of a sigh really when you see it pop up.”*
P44 C2 Prompt 30


*“I think that the 60 min duration was good for me I don’t think I would’ve liked a shorter duration the 30 min one”*
P05 C1 Prompt 60


*“I always wanted to try and engage and get as close to 100% as I could. Then I sort of felt a bit pressured in a way, I almost found it quite stressful […] trying to achieve [the breaks]. There was like an added stress that I didn’t really need”*
P51 C1 Prompt 60

A positive shift in workplace culture surrounding breaking up sedentary time was reported. However, perceived organizational and interpersonal expectations surrounding acceptable working behaviors was also cited as a factor which could negate adherence to postural breaks over time. 


*“The danger of [frequently breaking up sedentary time at work] though is that people might feel conscious that if they were, I was spending too long too long away from their desks they might be seen as a slacker for not working.”*
P55 C1 Prompt 30


*“I think you tend to think you’re being paid to work. So I focus on the work all the time without thinking about getting up […] I’m there to work.”*
P49 C2 Prompt 30

Overall, focus group participants perceived the prompts as an acceptable strategy to remind them to frequently break up their sedentary time at work. The standardized frequency of the prompts (30 or 60 min) and manual engagement with the application to report breaks, however, were not always convenient or suitable to the individual working patterns of participants within the trial. Focus group participants suggested linking prompt technology with a wearable device such as a wrist worn watch to allow automatic detection of breaks and physical activity, and tailoring strategies based on individual working patterns may increase adherence to future interventions over time.


*“I think it would have been a better study, if it connected to something like a watch, where it did actually monitor whether or not you were getting up”*
P05 C1 Prompt 60

Indeed, a common perception was that use of the smartphone application alone was not effective for establishing a habit to prompt frequent breaks to sedentary time over the 12 week intervention period.


*“[The intervention] wasn’t habit forming no for me no”*
P55 C1 Prompt 30

Consequently, the majority of participants in the focus groups felt their adherence to the prompts diminished over the 12 weeks, with the majority not continuing to use the application to remind them to break up their sedentary time since the intervention had ended. Participants reported that their diminished engagement over time was associated with a lack of incentives and a lack of physical or psychological reward in addition to workload pressures. Focus group participants identified strategies such as gamification, workplace challenges, educational messages and organizational engagement to compliment the smartphone application and encourage engagement over time.


*“Well, there’s [a company initiative], it’s more about general environment but something like that where possibly some built in rewards”*
P49 C2 Prompt 30


*“I think if I thought ‘oh okay I feel much better for that’ I would have carried on doing it, but it didn’t seem to make any difference to how I felt physically or emotionally. So I was getting nothing from [the intervention]. I wasn’t even getting emotional reward about saying ‘Oh well done you’ve worked hard now you get a break’ no you get nothing. […] it sounds ridiculous, but you feel like I want something or you gather coins or points or you have a competition between yourselves about how much you’ve earned […] or some facts that you’ve learned about something, average concentration time is 40 min, so it’s a good time, you’ll be working better when you sit down or just something”*
P55 C1 Prompt 30

## 4. Discussion

The aim of this study was to ascertain the contextual factors that moderate adherence to a smartphone application to break up prolonged SB in office workers. Overall, there were no differences between intervention arms relating to the types of working tasks and typical location throughout the trial which were predominately desk based. Most participants (72%) worked in a shared office and were therefore commonly not alone when prompted to break up their sedentary time. Between groups, a higher proportion of Prompt 60 participants were with a colleague compared to Prompt 30 participants over the 12 weeks. Indeed, focus group participants across both arms reported that peer support was an important factor for encouraging initial participation and ongoing engagement throughout the trial. Enhanced peer support was ascribed to an increased health awareness among colleagues, resulting in subtle, but noticeable, shifts in working cultures. This facilitated actions such as breaking up sedentary time during informal meetings and standing while on video calls to be more commonly perceived as acceptable working behaviors, with increased health driven motivation and the prompts themselves as a driver for compliance over time. Conversely, ecological momentary assessment (EMA) and focus group findings indicate that high workload (particularly relating to tasks requiring high levels of concentration and focus), meetings and being away from the workstation, in addition to perceived organizational expectations, had a detrimental impact on participant’s engagement with the prompts.

Our EMA findings indicate that while there were no apparent differences in the types of task between groups, there was a significant difference between the number of Prompt 60 participants who were with a colleague when prompted by the smartphone application. Taken together with our objective findings which indicate Prompt 60 participants yielded greater reductions in their total sedentary time (−69.6 min/8 h workday) than Prompt 30 participants (−37.0 min/8 h workday) [27], these contextual factors indicate that interpersonal support was a key driver for engagement and ongoing participation in the intervention. This may, in part, be attributed to the in-built social support network afforded by shared workspace or particular job roles of participants on this study. Indeed, increased social support has previously been recognized as an important factor for enhancing motivation while reducing the individual effort required to make positive behavioral changes [12]. Participants reported that enhanced awareness of sedentary occupational behaviors among their colleagues, either through participation itself, or through increased workplace conversations surrounding SB, helped to promote a subtle shift in working practices throughout the 12 weeks. It is unclear what interpersonal factors mediated this relationship, although previous work among desk-based employees within the call center environment highlighted the importance of seeing others perform the desired behavior [15,41]. It is therefore anticipated that recognized behavior change techniques such as modelling, peer support and social reward/comparison were inherent in these workplace settings and helped to reinforce the acceptability of these behaviors throughout the trial [42]. In addition to the e-health intervention strategy, it seems promoting e-health smartphone application use more widely among peers and increasing knowledge of health behaviors among co-workers, particularly within shared offices, is an important factor for shifting workplace perceptions around acceptable working behaviors, at least in the short-term. More research is, however, warranted to establish how changes and cultural adaptations may be maintained, and what influence this may have overall on behavioral outcomes given the typical decline in e-health usage over time [12,20]. 

Notably, both intervention groups felt their adherence to the e-health smartphone prompts diminished over the 12 weeks despite elements of social support and encouragement, which is of concern since intervention dose is associated with intervention efficacy [20] and indicates the limited sustainability of an e-health smartphone intervention among this population [20,43]. Over the 12 week intervention, EMA and focus group data indicated that high workload had a negative impact on participant’s ability to engage with the prompts, with participants in both Prompt 30 and Prompt 60 arms commonly and consistently reporting workload as a factor negatively influencing adherence to breaks. This finding supports previous qualitative research where high productivity demands and workload negate engagement in intervention strategies [44,45]. Crucially, our intervention message, while aligned to the current recommendations to reduce total and prolonged sedentary time and increase PA [46], simply encouraged participants to break up their sedentary time at regular intervals throughout the day. Evidently, as findings from the present study and wider literature demonstrate, embedding frequent transitions from sedentary to standing or ambulatory postures during working hours requires further education, awareness and opportunities built into already busy workloads. As these workload barriers appear to persist over time, this suggests that short-term feasibility trials could be developed to explore, co-design and refine possible work load and organizational intervention strategies. This would allow more rapid advances in our understanding of how to effectively target influential organizational factors around sedentary working practices before scaling up to larger scale or longer-term trials. Importantly, however, while increasing organizational awareness related to common workload issues and the association between high daily sedentary time, health and well-being outcomes may be an important factor for enhancing organizational support and optimizing longer-term user engagement, organizational strategies may in fact take longer to embed into organizational practice than environmental or multi-component interventions [47].

In addition to workload factors, the method for delivering the prompts via an e-health smartphone application was not always deemed to be feasible, particularly during client or student facing meetings when engaging with a smartphone was perceived as unprofessional or inappropriate. Focus group participants reported that it was not always practical or feasible to have their mobile phone in close proximity during working hours due to specific working tasks which meant that the participant was away from their workstation throughout the day. This unique insight into the contextual factors influencing adherence to an e-health workplace intervention, in conjunction with behavioral findings [27], suggests that smartphone prompts are more likely to influence total and prolonged sedentary time when an individual is engaged in desk-based working tasks. However, smartphone prompts and reminders may not always be practical or suitable depending on work task and location. For example, when in an informal meeting, cultural factors such as increased employee awareness appear to be more influential for shifting employee perceptions and behaviors around typical sedentary working postures.

The standardized break frequency (set at 30 or 60 min intervals) was informed by the high proportion of desk-based workers who accrue their daily worktime SB in prolonged bouts of either 30 or 55 min [3] and the increasing evidence demonstrating a detrimental association between prolonged sedentary bouts and health, well-being and work related outcomes [5,6,7,48]. Delivering standardized break frequencies was not always perceived as appropriate or feasible for participants, with focus group participants reporting the 60 min prompt to be more feasible during work compared to 30 min prompts which were often found to be disruptive to daily working pattern and concentration. Similar to another computer-based e-health intervention [49], the manual entry required to log a break via the application was also commonly perceived as clunky and inconvenient, and there were calls from participants to map the smartphone technology to inbuilt accelerometer capabilities to automatically detect when a person had broken up their sedentary time; link with existing wearable technologies such as a wrist worn device; or sync with participants workday calendar/schedule to tailor the intervention strategy and detect when participants were in meetings and avoid prompts at inconvenient or inappropriate times. While tailoring future strategies to individual needs seems important, a cross sectional study among predominately desk-based workers also showed that employees are naturally quicker at transitioning between seated and standing postures later on in the working day [50]. Considering the temporal dynamics of SB within workplace in addition to the method and frequency of prompt delivery may also help to optimize compliance and behavior change and increase acceptability and feasibility over time.

Without the addition of environmental adaptations such as height-adjustable workstations, participants in the present trial described a number of strategies to achieve a break in their workday sedentary time when prompted. These involved simply alternating between a seated and standing posture at their workstation, standing during informal team meetings, standing when on a call or in a conference meeting and reports of increasing purposeful activity, i.e., taking a walk when prompted. Behavioral findings from this intervention indicate that increasing purposeful activity during working hours was not a common strategy across the sample as there was no indication that changes in ambulation or physical activity occurred [27]. Encouragingly, however, the present intervention resulted in reductions of ≥30 min in total sedentary time [27], which may be clinically meaningful for health [51,52] and presents employers with a potentially cost effective strategy to promote the use of hourly prompts to reduce their employees total and prolonged work time SB [53]. Proposed strategies to enhance engagement included the development of additional components to optimize behavioral changes among desk-based workers through incorporating physical or psychological incentives, i.e., gamification, workplace challenges and educational messages. This multi-component approach is widely advocated within the wider literature due to the effectiveness for reducing total and prolonged sedentary time [12,51] compared to single level interventions. Complex interventions, however, are typically more costly to deliver, and businesses need to justify a return on investment against health or productivity markers. Future interventions should be mindful of this and seek to build a business case alongside intervention development, delivery and evaluation to simultaneously address individual and business needs. 

Overall, our findings indicate that the e-health smartphone application may work as a tool to increase understanding and awareness of sedentary behaviors among desk-based employees. However, as highlighted, the e-health intervention should be refined based on qualitative and contextual feedback to optimize participant engagement over time. Importantly, given the common barriers to adherence appeared consistent between groups throughout the trial, overcoming these in the early stages of a future intervention may be an important factor for enhancing intervention compliance and sustainability. Our preliminary findings warrant further investigation to determine whether an e-health smartphone application may support behavioral changes as a stand-alone or part of a multi-component workplace intervention.

### Strengths and Limitations

A strength of the present trial was that it was delivered in real-world workplace settings which enhances its ecological validity. Participant attrition was 7% and 14% at 6 and 12 weeks, respectively, which is lower than experienced within other workplace intervention trials [12] and there were no intervention withdrawals due to adverse events [14]. Our findings provide novel, contextual data which may help to improve participant engagement and can inform the development of future e-health trials among desk-based office workers to further explore effectiveness and sustainability over time. The e-health smartphone application on iOS platform was not able to capture participant adherence and application usage over the trial period, it is likely that qualitative reports of diminished use and engagement across the intervention had a detrimental impact on the preliminary effectiveness of the intervention on behavioral outcomes [20]. Due to the number of categorical variables and low participant responses to the EMA prompt and review forms, some outcomes violated the assumption of chi-square statistical test that each variable should have an expected count >5.

Participant fidelity to the EMA measures demonstrated, on average, over one-third of valid responses over the 12 week trial. This compliance has been deemed acceptable within clinical populations evaluating mental health [29], although it was lower than the threshold (75%) considered when used as a surrogate measure for monitoring sedentary time, standing and stepping behavior at work [54]. Guidance on the duration of EMA sampling recommends up to a maximum of 6 days which was exceeded over the duration of this 12 week trial. This prolonged EMA measurement period may have negatively impacted participant compliance over time [29]. EMA sampling should therefore be considered in future trials to capture data over a monitoring period of up to 6 days at intervals over the duration of the trial to optimize participant compliance to this real-time, contextual feedback.

## 5. Conclusions

In conclusion, initial findings from this e-health smartphone-delivered feasibility trial indicate that when working on a sedentary working task and located at an individual workstation, hourly prompts may be more acceptable and feasible for promoting a reduction in total and prolonged sedentary time than 30 min prompts. Furthermore, interpersonal support appears important for promoting subtle shifts in daily working practices, with calls to increase organizational support to better understand the link between high workload, sedentary time, and the potential impact on health, well-being and productivity. Further research is warranted to optimize the method of e-health intervention delivery over time and tailor intervention strategies to meet the needs of individuals in the workplace prior to delivering a large-scale intervention. Given the common barriers to adherence were consistent between groups throughout the trial, overcoming these in the early stages of a future intervention may be an important factor for enhancing intervention compliance and sustainability.

## Figures and Tables

**Table 1 ijerph-18-09627-t001:** Pooled chi-squared interaction response between mood (affect) and company (alone or with colleagues) across the 12 week intervention.

	Prompt 30		Prompt 60	
Affect	Alone	With Others	Alone	With Others
Low negative	3.6 (18)	2.4 (12)	2.9 (14)	6.9 (34)
Pleasantness	23.7 (119)	24.1 (121)	9.2 (45)	27.1 (133)
High positive	0.6 (3)	2.8 (14)	0.2 (1)	2.0 (10)
Strong engagement	5.8 (29)	4.6 (23)	2.2 (11)	8.8 (43)
High negative	5.4 (27)	6.4 (32)	3.9 (19)	10.4 (51)
Unpleasantness	2.2 (11)	2.2 (11)	2.4 (12)	1.2 (6)
Low positive	18.3 (20)	9.7 (49)	8.1 (40)	11.0 (54)
Disengagement	0.6 (3)	0.2 (1)	0.6 (3)	1.4 (7)

Data are presented as the mean percentage % (*n*) by intervention arm.

**Table 2 ijerph-18-09627-t002:** Weekly review responses according to intervention arm.

	Prompt 30	Prompt 60	U (*z*)	*p*
How much did you break up your sedentary time this week compared to last week?	7.0	6.0	2322.5 (−0.9)	0.359
How often did you engage with the prompts this week?	7.0	7.0	2440.5 (−0.4)	0.665
How likely are you to engage with the prompts next week?	8.0	8.0	2779.5 (1.0)	0.336
How helpful do you find the prompts?	8.0	7.5	2425.5 (−0.5)	0.620

Data are presented as the median Likert scores, Mann–Whitney U statistic (U), z-score (*z*) and significance (*p*).

## Data Availability

The data presented in this study are openly available in Lancaster University Repository [PURE] at [https://pure.lancs.ac.uk/files/335350872/LU_PURE_data_sheet_Rise_and_Recharge.V2.xlsx] (accessed on 9 September 21).

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
