# Peer review of "Rise and Recharge: Exploring Employee Perceptions of and Contextual Factors Influencing an Individual-Level E-Health Smartphone Intervention to Reduce Office Workers’ Sedentary Time at Work"

_ijerph, 2021, doi:10.3390/ijerph18189627_

Round 1
Reviewer 1 Report
In the manuscript entitled “Rise and Recharge: Exploring employee perceptions of and contextual factors influencing an individual-level e-health smartphone intervention to reduce office workers’ sedentary time at work” authors aim to investigate the contextual factors that influence the adherence to an e-health smartphone-based intervention targeting sedentary time during waking hours in office workers. Authors assessed qualitative and quantitative methods and found that e-health intervention with prompts delivered by smartphone is effective in reducing sitting time in sedentary workers. In particular, hourly prompts may be more acceptable and feasible for than 30-minute prompts. Furthermore, peer support was an important endorsement in engaging the trial, while high workload, meetings and being away from the workstation had a negative impact.
This preliminary feasibility work is well conducted and well written. Conclusions are supported by results and may help future research in developing and test new strategies to improve active lifestyle and to reduce sedentary behaviour during work time.
The only suggestion from me is that Authors should specify if clinical characteristics were different between groups (BMI, hypertension etc..).
Author Response
Thank you for your feedback and comments on the manuscript. Our previously published paper (Morris et al., 2020) presented the behavioural and cardiometabolic data for all participants in this intervention. As this data is already published and publicly available via open access, we do not want to duplicate this data in this paper. Nonetheless, in the discussion section we discuss the contextual findings in relation to the behavioural and cardiometabolic outcomes (page 9, line 386-392) align both papers, whilst still focussing on the aim of this paper, which was to present the contextual findings rather than any cardiometabolic or behavioural data.
Morris, A. S., et al. (2020). "Rise and Recharge: Effects on Activity Outcomes of an e-Health Smartphone Intervention to Reduce Office Workers’ Sitting Time." International Journal of Environmental Research and Public Health 17(24): 18.
Reviewer 2 Report
General comment
Thank you for submitting a research article on this very important topic. It is extremely important to explore the contextual factors that influence the adherence to an e-health smartphone-based intervention targeting total and prolonged sedentary time during working hours among office workers. This work allows other researchers and practitioners to come up with creative ways to increase adherence to interventions aimed at reducing sedentary time.
Overall, the manuscript is well written, and the method is well-justified.
Specific comments
Point 1. Page 1 – Line 6, affiliation should start with ascending order, e.g., Abigail S. Morris affiliation starts at 2 and should start at 1.
Point 2. Page 1 – Line 8-19, I suggest not writing the full name of the authors, just email and the abbreviation of the author's name in parentheses.
Point 3. Page 1 – Line 25-26, acronyms/abbreviations should be defined the first time they appear in each of three sections: the abstract; the main text; the first figure or table. When defined for the first time, the acronym/abbreviation should be added in parentheses after the written-out form. Please, consider adding the acronym EMA in parentheses after the written-out form.
Point 4. Page 2 – Line 52-54, please, consider rephrasing the sentence with more information, e.g., add if it is results of short term follow up (up to three months) or medium-term follow-up (3 to 12 months); also, the intervention with the sit-stand table was applied alone or in combination with other strategy, such as counselling. Are the values presented right? Please take a look at the values presented in the systematic review. Also, it looks like the numbers should be negative.
Point 5. Page 4 – Line 151-159, the reference 34 is not being used in the main text.
Point 6. Page 4 – Line 168, consider writing “Consolidated criteria for reporting qualitative research” and then add the acronym COREQ in parentheses after the written-out form.
Point 7. Page 4 – Line 175-176, I have two suggestions at this point. First suggestion, rewriting this paragraph and include more information about the ANOVA model (e.g., within-weeks and between-groups and how many levels) in order to facilitate understanding. Second suggestion, as the Likert data were nonnormally distributed, consider using a nonparametric test rather than an ANOVA. For example, Friedman’s nonparametric tests could be used for within-subject differences between weeks in each of the 2 groups separately.
Point 8. Page 4 – Line 182-184, please, check the Prompt 60 percentage. If this group did not have any dropout during the trial the value should be 100%.
Point 9. Page 5 – Line 195-197, please, consider citing your study here, reference 27.
Point 10. Page 6 – Line 237-239, which test was used in this analysis? According to the section 2.4.3. Statistical analysis, an ANOVA, right? However, X2 is presented instead of F-value followed by degree of freedom and p-value. Furthermore, if these data are nonnormally distributed a nonparametric test should be used.
Point 11. Page 6 – Line 242 – section 3.1.1. Focus group, exactly how many participants were in the focus group, 8 or 10? The text in line 245 says that 10 participants participated in the focus group, five participants from Company 1 and five participants from Company 2, please check. In the section, it is possible to count 9 codes (P04 C1 Prompt 30, P05 C1 Prompt 30, P55 C1 Prompt 30, P51 C1 Prompt 60, P39 C2 Prompt 30, P42 C2 Prompt 30, P44 C2 Prompt 30, P48 C2 Prompt 30, and P49 C2 Prompt 30), four from company 1 and five from company 2. Also, there are some unique codes that are different in the text that have not been counted, see below.
Is this unique code - "P05 C1 Prompt 30" – correct on page 7 line 314? Because on page 7 line 325 and page 8 line 335 it appears as "P05 C1 Prompt 60", and then on page 8 line 357 it goes back to appearing as "P05 C1 Prompt 30".
Is this unique code - "P51 C1 Prompt 30" – correct on page 8 line 338? Because on page 6 line 259 and page 7 line 286 it appears as "P51 C1 Prompt 60".
Is this unique code - "P55 C2 Prompt 30" – correct on page 8 line 379? Because on page 6 line 267, page 6 line 275, page 7 line 284, page 7 line 295, page 8 line 345, and page 8 line 361 it appears as "P55 C1 Prompt 30".
Point 12. Page 12 – Line 564, the names of the supplementary material should be “Image S1: …”, “Text S1: …”, “Table S1: …”, “Figure S1: …”, “Figure S2: …”.
Point 13. Page 12, some information is missing, such as “Institutional Review Board Statement”, “Informed Consent Statement”, and “Data Availability Statement”. Please provide details regarding where data supporting reported results can be found, including links to publicly archived datasets analyzed or generated during the study. Please refer to suggested Data Availability Statements in section “MDPI Research Data Policies” at https://www.mdpi.com/ethics.
Point 14. Page 12 – Line 581, consider checking all references (authors name, page, issue, and volume). Also, include the digital object identifier (DOI) for all references where available.

Author Response
We would like to thank the reviewer for this kind feedback and for taking the time to review the manuscript in detail. Please see our responses to your comments and suggestions below, which we feel have helped to strengthen this manuscript.
Specific comments
Point 1. Page 1 – Line 6, affiliation should start with ascending order, e.g., Abigail S. Morris affiliation starts at 2 and should start at 1.
- Thank you for highlighting this oversight. The first author has a dual affiliation, in which the first affiliation was missed – this has now been amended
Point 2. Page 1 – Line 8-19, I suggest not writing the full name of the authors, just email and the abbreviation of the author's name in parentheses.
- Thank you. The authors full names have been removed from the affiliations list and only their email and initials are presented.
Point 3. Page 1 – Line 25-26, acronyms/abbreviations should be defined the first time they appear in each of three sections: the abstract; the main text; the first figure or table. When defined for the first time, the acronym/abbreviation should be added in parentheses after the written-out form. Please, consider adding the acronym EMA in parentheses after the written-out form.
- Thank you for highlighting this oversight. EMA has now been included in parentheses in the abstract.
Point 4. Page 2 – Line 52-54, please, consider rephrasing the sentence with more information, e.g., add if it is results of short term follow up (up to three months) or medium-term follow-up (3 to 12 months); also, the intervention with the sit-stand table was applied alone or in combination with other strategy, such as counselling. Are the values presented right? Please take a look at the values presented in the systematic review. Also, it looks like the numbers should be negative.
- Apologies for the lack of clarity which we have now integrated. The section now reads as follows (line 51-55):
Workplace interventions involving height-adjustable workstations, with or without the addition of information and counselling, have been shown to be effective for reducing total and prolonged occupational sedentary time among desk-based workers over three months (- 84 to - 116 minutes per workday (Shrestha, Kukkonen‐Harjula et al. 2018)), with observed reductions maintained over 12-month follow-ups (Healy, Eakin et al. 2016, Edwardson, Yates et al. 2018).
Point 5. Page 4 – Line 151-159, the reference 34 is not being used in the main text.
- Thank you for highlighting this omission; reference 34 has since been added within text and updated accordingly in the reference list.
Point 6. Page 4 – Line 168, consider writing “Consolidated criteria for reporting qualitative research” and then add the acronym COREQ in parentheses after the written-out form.
- Thank you – we have now amended this.
Point 7. Page 4 – Line 175-176, I have two suggestions at this point. First suggestion, rewriting this paragraph and include more information about the ANOVA model (e.g., within-weeks and between-groups and how many levels) in order to facilitate understanding. Second suggestion, as the Likert data were nonnormally distributed, consider using a nonparametric test rather than an ANOVA. For example, Friedman’s nonparametric tests could be used for within-subject differences between weeks in each of the 2 groups separately.
- Thank you for this comment. We have rewritten this paragraph to facilitate understanding. Please our response to Point 10 (below) in relation to our justification for the analysis.
Point 8. Page 4 – Line 182-184, please, check the Prompt 60 percentage. If this group did not have any dropout during the trial the value should be 100%.
- Thank you for highlighting this. This has been amended to 100%.
Point 9. Page 5 – Line 195-197, please, consider citing your study here, reference 27.
- Thank you for this suggestion. We spent some time considering this but feel that it’s important to keep the results section focussed on the present study findings surrounding the contextual outcomes of the intervention. However, these results have been discussed in relation to our previous paper and the behavioural findings which appear to corroborate these contextual findings [lines 386-392].
“Our EMA findings indicate that while there were no apparent differences in the types of task between groups, there was a significant difference between the number of Prompt 60 participants who were with a colleague when prompted by the smartphone application. Taken together with our objective findings which indicate Prompt 60 participants yielded greater reductions in their total sedentary time (-69.6 min/8 h workday) than Prompt 30 participants (-37.0 min/8 h workday) [27], these contextual factors indicate that interpersonal support was a key driver for engagement and ongoing participation in the intervention.”
Point 10. Page 6 – Line 237-239, which test was used in this analysis? According to the section 2.4.3. Statistical analysis, an ANOVA, right? However, X2 is presented instead of F-value followed by degree of freedom and p-value. Furthermore, if these data are nonnormally distributed a nonparametric test should be used.
- Thank you for this comment. On reflection we feel it was important to make a clarification within the writing regarding the analysis that was conducted. We used chi square analysis for the nominal/categorical data within the prompt and review questionnaires, which has remained unchanged. However, we felt this needed to be written more clearly within the text (see revisions below). We also re-ran the non-parametric analysis for the Likert data using a Mann-Whitney U test to determine whether there were differences between the two groups. We did consider using the Friedmans test, however the data violated some of the key assumptions for this test and therefore the Mann-Whitney U was the best fit. Please see updated text on page 4 (line 160-166):
“Descriptive statistics were conducted on the complete EMA dataset. Due to the categorical nature of the Prompt and Review data, Chi-square tests were conducted using SPSS (IBM SPSS Statistics 26, USA) to explore associations between participant adherence and influencing contextual factors, with alpha set to p ≤ 0.05 (Field 2010). Furthermore, based on a complete case analysis including all data collected across the 12 weeks, Likert scales from the weekly review forms were analyzed using the Mann-Whitney U test to determine differences between Prompt 30 and Prompt 60 arms (Field 2010).”
- Table 2 (page 6, line 255-226) has also been updated to reflect this analysis and the corresponding text on page 5 now reads as follows (line 214-219):
“Results from the weekly review questionnaire Likert data are presented in Table 2. Likert data were not normally distributed, and therefore non-parametric analyses were conducted. Overall, distributions of Likert scores for Prompt 30 and Prompt 60 arms were similar, and medians scores across groups were not statistically significant (Dinneen and Blakesley 1973).
Descriptive statistics indicated that the main factors negatively influencing participant engagement in the prompts included high workload (27.3%), meetings (18.9%) and being away from their desk or mobile (10.5%). Conversely, the e-health smartphone prompts (11.3%) and an intrinsic health-driven motivation (11.9%) were reported as positive factors influencing adherence. Chi square analysis of the categorical data indicated that there were no statistically significant associations between interventions arms and review questionnaire responses across the 12-week trial [ꓫ2 range p = .82 - .858].”
Point 11. Page 6 – Line 242 – section 3.1.1. Focus group, exactly how many participants were in the focus group, 8 or 10? The text in line 245 says that 10 participants participated in the focus group, five participants from Company 1 and five participants from Company 2, please check. In the section, it is possible to count 9 codes (P04 C1 Prompt 30, P05 C1 Prompt 30, P55 C1 Prompt 30, P51 C1 Prompt 60, P39 C2 Prompt 30, P42 C2 Prompt 30, P44 C2 Prompt 30, P48 C2 Prompt 30, and P49 C2 Prompt 30), four from company 1 and five from company 2. Also, there are some unique codes that are different in the text that have not been counted, see below.
- Ten participants took part in the focus groups overall. We have provided full details in the consort flow chart (by intervention arm), which we appreciate may be a little confusing/ misleading in the manuscript. The following text has therefore been amended in the manuscript to mirror the consort flow chart and enhance the overall clarity for the reader (line 228- 233):
“Verbatim quotations are presented as participant number (P1-56), company (Company 1 [C1] or Company 2 [C2]) and intervention arm (Prompt 30 or Prompt 60; e.g. P06 C1 Prompt 30). Ten participants, five (n = 4 male) from C1 and five (n = 1 male) from C2, took part. Overall there were eight Prompt 30 participants and two Prompt 60 participants (Figure S1: Consort flow diagram of enrolment, allocation, follow up and analysis) with the mean focus group duration being 40.6 ± 2.0 minutes.”
- In relation to quotations only being provided by nine participants; the quotations have been identified to best reflect the theme being presented and there was no intention to present a quote from each participant. To the authors knowledge, it is not typical/required to present quotations from each participant when using thematic analysis. We believe that the quotations provided have helped to demonstrate the contextual factors outlined by the research and therefore feel that this is the best way to present these qualitative results.
Is this unique code - "P05 C1 Prompt 30" – correct on page 7 line 314? Because on page 7 line 325 and page 8 line 335 it appears as "P05 C1 Prompt 60",
- Thank you for highlighting the inconsistencies and typos within this section. Focus group participants have been cross checked again against the data collection dataset and these have been corrected throughout. In this specific case, it was P05 C1 Prompt 60.
Then on page 8 line 357 it goes back to appearing as "P05 C1 Prompt 30".
- We have checked this quotation which has been provided by P55 C2 Prompt 30 and ensured it is correct. Other edits (see responses below), have been revised to address this and ensure consistency.
Is this unique code - "P51 C1 Prompt 30" – correct on page 8 line 338? Because on page 6 line 259 and page 7 line 286 it appears as "P51 C1 Prompt 60".
- Thank you for the attention to detail and highlighting this error. This has been corrected to P51 C1 Prompt 60.
Is this unique code - "P55 C2 Prompt 30" – correct on page 8 line 379? Because on page 6 line 267, page 6 line 275, page 7 line 284, page 7 line 295, page 8 line 345, and page 8 line 361 it appears as "P55 C1 Prompt 30".
- Thank you for the attention to detail and highlighting this error. This has been corrected to P55 C1 Prompt 30.
Point 12. Page 12 – Line 564, the names of the supplementary material should be “Image S1: …”, “Text S1: …”, “Table S1: …”, “Figure S1: …”, “Figure S2: …”.
- Thank you for highlighting this formatting error. Labelling has been amended in text and in the supplementary file to reflect these changes.
Point 13. Page 12, some information is missing, such as “Institutional Review Board Statement”, “Informed Consent Statement”, and “Data Availability Statement”. Please provide details regarding where data supporting reported results can be found, including links to publicly archived datasets analyzed or generated during the study. Please refer to suggested Data Availability Statements in section “MDPI Research Data Policies” at https://www.mdpi.com/ethics
- Please accept our apologies for this omission. The following text and information have now been added to the manuscript [Lines 550-556].
“Institutional review board statement: The study was conducted according to the guidelines of the Declaration of Helsinki, and approved by the Institutional Review Board (or Ethics Committee) of the College of Engineering, Swansea University (protocol code 2018-147, date of approval November 2018).”
“Informed consent statement: Informed consent was obtained from all participants involved in the study.”
“Data availability statement: The data presented in this study are openly available in Lancaster University Repository [PURE] at [https://doi.org/10.17635/lancaster/researchdata/].”
Point 14. Page 12 – Line 581, consider checking all references (authors name, page, issue, and volume). Also, include the digital object identifier (DOI) for all references where available.
- All references have been checked and, where applicable, DOI’s have been added. This has not been highlighted on tracked changes due to the use of referencing (EndNote) software.